# SPIKING GRAPH PREDICTIVE CODING

## ABSTRACT

Graph Neural Networks (GNNs) have shown strong performance on structured data, but they often suffer from poor calibration and limited generalisation under out-of-distribution (OOD) data, which poses a critical challenge for trustworthy graph learning in real-world applications. In contrast, spiking neural networks (SNNs) and predictive coding (PC) provide biologically grounded mechanisms for event-driven computation and local error correction, which naturally promote robustness and calibrated uncertainty. Inspired by these principles, we propose SpIking GrapH predicTive coding (SIGHT), a framework that integrates PC dynamics with spiking computation for graph learning. SIGHT preserves the architectural flexibility of modern GNNs while replacing global backpropagation with local, spike-driven error correction, yielding learning dynamics that are inherently robust to distribution shifts. Experiments on five graph datasets with two types of OOD scenarios show that SIGHT delivers competitive predictive accuracy, better-calibrated uncertainty, and stronger OOD detection than standard GNNs. Beyond accuracy, the error-driven spiking dynamics provide natural explanations for uncertainty, and the event-driven computation makes SIGHT attractive for deployment on power-constrained hardware, highlighting its potential as a principled and efficient alternative for robust graph learning. Source code is available at: https://anonymous.4open.science/r/SGPC_ood-B9E4.

## 1 INTRODUCTION

Graph Neural Networks (GNNs) have demonstrated strong performance across a wide range of safety-critical and high-stakes applications, including domains such as healthcare, finance, transportation, and cybersecurity, where reliable decision-making under uncertainty is of paramount importance (Xia et al., 2021; Liu et al., 2022; Khoshraftar & An, 2024; Han et al., 2025; Chen & Li, 2025). However, conventional GNNs often exhibit poor uncertainty calibration and degraded out-of-distribution (OOD) generalisation when the test distribution deviates from the training distribution (Wu et al., 2022; Yang et al., 2022; Wu et al., 2024; Wang et al., 2025; Li et al., 2025a). These shortcomings can lead to critical failures in practice. For example, a GNN used for drug discovery may output high confidence in a molecule with unseen chemical structures, while in autonomous navigation, a perception model may misclassify rare obstacles with undue certainty. Such risks underscore why unreliable uncertainty estimation hinders the deployment of GNNs in safety-critical applications where trustworthy confidence assessments are indispensable (Kaur et al., 2022; Liang et al., 2022; Fuchsgruber et al., 2025).

To mitigate these risks, several approaches have recently been proposed to explicitly address the non-IID characteristics of node classification tasks (Hsu et al., 2022; Wang et al., 2021). While effective for aligning predicted probabilities with empirical correctness on in-distribution (ID) data, these methods often fail to transfer calibration gains to shifted distributions (Ovadia et al., 2019; Wiles et al., 2022). This highlights the need for training strategies that endow models with stronger intrinsic uncertainty estimation, yielding more reliable ID and OOD estimates out of the box while still allowing further refinement through post-hoc calibration (Trivedi et al., 2024).

The recently proposed Graph Predictive Coding Networks (GPCN) (Byiringiro et al., 2022) show improved calibration on node classification tasks and robustness against adversarial attacks by learning through local error feedback and iterative inference of predictive coding (PC) (Rao & Ballard, 1999). Besides offering a scalable alternative to backpropagation, PC aligns with biological principles of neural processing, making it a compelling foundation for uncertainty estimation in graph

learning. However, directly integrating PC with traditional GNNs is challenging, as conventional GNNs rely on dense, synchronous updates that conflict with the iterative and event-driven dynamics of PC. To address this limitation, we introduce spiking neural networks (SNNs) (Rathi et al., 2023; Yin et al., 2024), enabling graph predictive coding to operate with sparse and asynchronous spikes that align more naturally with local error correction and the biologically inspired principles of predictive coding.

**Contributions.** In summary, this work makes the following contributions:

- We propose SpIking GrapH predicTive coding (**SIGHT**), a biologically plausible paradigm that replaces global backpropagation with local Hebbian-style learning to improve OOD uncertainty estimation. Its predictive coding strategies capture mismatches between predictions and observations, naturally explaining model confidence.

- We present a theoretical analysis that establishes the convergence dynamics and uncertainty quantification properties of SIGHT. These results offer formal guarantees that support model robustness and explain its improved generalisation under distribution shifts.

- Experiments show that SIGHT achieves superior OOD generalisation and more calibrated uncertainty estimates than traditional GNNs across multiple benchmarks. Moreover, SIGHT complements post-hoc calibration models, delivering consistent improvements with both GCN and GAT backbones.

## 2 RELATED WORK

**OOD Generalisation and Uncertainty Estimation in Graphs.** OOD generalisation and uncertainty estimation have gained increasing attention in graph representation learning, as real-world graph data often exhibit significant distribution shifts across tasks or domains (Bazhenov et al., 2023; Liu et al., 2023a;b; Li et al., 2025b). These shifts can arise from changes in node features, graph structure, or even concept drift that alters the label–feature correspondence, all of which may severely degrade model reliability. Traditional GNNs, however, remain vulnerable to such shifts because they are typically optimised to fit the training distribution and lack mechanisms to quantify or adapt to uncertainty (Yehudai et al., 2021; Wu et al., 2022; Yang et al., 2022; Yu et al., 2023; Yuan et al., 2025). As a result, their predictions tend to be overconfident and poorly calibrated when deployed on unseen distributions, limiting their applicability in safety-critical scenarios such as drug discovery, fraud detection, and recommender systems.

**Model Calibration in Graphs.** Existing approaches largely fall into two categories: post-hoc calibration (Guo et al., 2017; Kull et al., 2019; Wang et al., 2021; Hsu et al., 2022), which adjusts confidence after training, and intrinsic enhancement techniques (Trivedi et al., 2024; Thiagarajan et al., 2022), which modify training to yield better-calibrated predictions. Post-hoc calibration models, such as temperature scaling (Guo et al., 2017), Bayesian graph models (Gal & Ghahramani, 2016), and deep ensembles (Lakshminarayanan et al., 2017), improve the alignment between predicted probabilities and empirical correctness by adjusting model confidence after training, but they often fail under distribution shifts. By contrast, Trivedi et al. (2024) propose an intrinsic method, G-$\Delta$UQ, which modifies the training process to produce calibrated predictions. However, these approaches fail to explain the sources of uncertainty, although such interpretability is vital for building trustworthy models. Moreover, current models lack the adaptability and energy efficiency characteristic of brain-like inference.

**Predictive Coding.** Predictive coding is a neurobiological theory proposing that the brain continually generates top-down predictions of sensory input, while bottom-up signals convey only the prediction errors, thereby driving perception and learning (Hebb, 1949; Rao & Ballard, 1999). This biologically inspired alternative to backpropagation (Friston, 2018; Salvatori et al., 2023) has been applied in generative modeling (Ororbia & Kifer, 2022), continual learning (Ororbia et al., 2020), reinforcement learning (Ororbia & Mali, 2023), graph learning (Byiringiro et al., 2022), and arbitrary network learning (Salvatori et al., 2022). Moreover, Byiringiro et al. (2022) find that replacing traditional BP in GNNs with PC can achieve better calibration than traditional GNNs in ID settings. However, predictive coding has not yet been explored in the context of OOD generalization and calibration, particularly when combined with spiking GNNs.

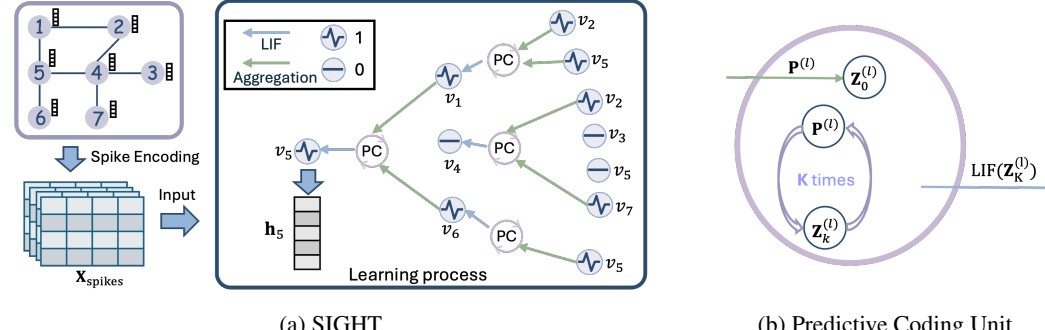

(a) SIGHT                                             (b) Predictive Coding Unit

Figure 1: (a) Overview of the proposed framework SIGHT. First, node features are encoded into spike trains $\mathbf{X}_{\text{spikes}}$ as input (Eq. 1). Then, the node representations are learned by aggregating and propagating neighbor information through graph convolution layers (Eq. 2). Before propagation, predictions are refined via an iterative predictive coding unit (Eq. 3). Over multiple steps, neuronal activities converge to stable representations, and synaptic weights are updated through Hebbian-style local learning. (b) Predictive coding unit, where residuals between predictions and latent states are converted into error spikes to iteratively update activations with Eq. 3 for $K$ times.

# 3 METHODOLOGY

In this paper, we propose **SIGHT**, a flexible framework for reliable uncertainty estimation in GNN-based classification, as shown in Figure 1. Further details on reproducibility, pseudocode, and computational complexity analysis are provided in Appendix B.

## 3.1 NOTATION

We consider a graph $\mathcal{G} = (\mathcal{V}, \mathcal{E})$ composed of a node set $\mathcal{V}$ and an edge set $\mathcal{E}$, where $|\mathcal{V}| = N$ and the graph is represented by an adjacency matrix $\mathbf{A} \in \mathbb{R}^{N \times N}$. Each node $v_i \in \mathcal{V}$ is associated with a feature vector $\mathbf{x}_i \in \mathbb{R}^F$, which together form the feature matrix $\mathbf{X} \in \mathbb{R}^{N \times F}$.

We adopt the symmetrically normalised adjacency matrix $\tilde{\mathbf{A}} = \mathbf{D}^{-\frac{1}{2}}(\mathbf{A} + \mathbf{I})\mathbf{D}^{-\frac{1}{2}}$, where $\mathbf{D}$ is the degree matrix. An $L$-layer SIGHT model maintains hidden dimensions $d_0 = F, d_1, \ldots, d_L$ and weight matrices $\mathbf{W}^{(l)} \in \mathbb{R}^{d_{l-1} \times d_l}$. We focus on node classification, where each node $v_i$ is assigned a label $y_i \in 1, \ldots, C$, and the labels of all nodes are collectively represented by a one-hot matrix $\mathbf{Y} \in 0, 1^{N \times C}$. In this paper, bold uppercase letters (e.g., $\mathbf{X}, \mathbf{W}, \mathbf{Z}$) denote matrices.

## 3.2 POISSON ENCODING OF NODE FEATURES

Given the node feature matrix $\mathbf{X} \in \mathbb{R}^{N \times F}$, we first normalise each feature channel to the range $[0, 1]$, yielding $\tilde{\mathbf{X}}$. Each scalar feature $\tilde{X}_{i,f}$ is then converted into a spike train using a Poisson process:

$$\mathbf{X}_{\text{spikes}}[t, i, f] \sim \text{Bernoulli}(\tilde{\mathbf{X}}_{i,f}), \quad \mathbf{X}_{\text{spikes}} \in \{0, 1\}^{T \times N \times F}. \tag{1}$$

At each simulation step $t$, the slice $\mathbf{X}_{\text{spikes}}[t]$ is used as the network input.

## 3.3 FORWARD PASS WITH PREDICTIVE CODING INFERENCE

To illustrate the method, we first present SIGHT with GCN (Kipf & Welling, 2017) as the backbone. However, the framework is not restricted to GCN and can be readily extended to other GNN architectures, such as GAT (Veličković et al., 2018), highlighting its broad applicability. The forward pass integrates graph convolution with predictive coding dynamics. For each layer $l \in \{1, \ldots, L\}$ and timestep $t$, the procedure consists of:

**Graph Convolution.** We compute latent predictions as:

$$\mathbf{P}^{(l)} = \tilde{\mathbf{A}}\mathbf{H}^{(l-1)}\mathbf{W}^{(l)}, \quad \mathbf{H}^{(0)} = \mathbf{X}_{\text{spikes}}[t]. \tag{2}$$

**Predictive Coding Loop.** We initialise the latent state $\mathbf{Z}_0^{(l)} \leftarrow \mathbf{P}^{(l)}$ and perform $K$ inference iterations:

$$\mathbf{E}_k^{(l)} = \mathbf{P}^{(l)} - \mathbf{Z}_k^{(l)}, \quad \mathbf{U}_k = \mathbf{P}^{(l)} + \gamma \, \mathrm{LIF}_{\mathrm{err}}(\mathbf{E}_k^{(l)}), \quad \mathbf{Z}_{k+1}^{(l)} = \mathrm{LIF}_{\mathrm{pred}}(\mathbf{U}_k). \tag{3}$$

Here, $\mathbf{Z}^{(l)} \in \mathbb{R}^{N \times d_l}$ denotes layerwise latent predictions, $\mathbf{P}^{(l)} \in \mathbb{R}^{N \times d_l}$ denotes instantaneous pre-activations, and $\mathbf{E}_k^{(l)}$ denotes the local residual, $\gamma$ is a correction gain, and $\mathrm{LIF}_{\mathrm{err}}, \mathrm{LIF}_{\mathrm{pred}}$ denote leaky integrate-and-fire dynamics.

After $K$ iterations, the final latent state $\mathbf{Z}_K^{(l)}$ serves as the layer output. The spike tensor $\mathbf{Z}_K^{(l)}$ is rectified and propagated to the next layer. At the final layer, the corrected spikes are passed to the classifier. This iterative procedure can also be interpreted from an energy-minimisation perspective, where the dynamics of LIF neurons approximate gradient descent on a local error functional, as described in Appendix B.2.

**Uncertainty Quantification via Error Statistics.** We define the predictive coding confidence at layer $l$ for node $i$ as $u_i^{(l)} = \exp\left(-|\mathbf{E}_{i,:}^{(l)}|_2^2\right)$, which approximates the likelihood of observation under Gaussian residual noise. This provides a principled mapping from error magnitudes to epistemic uncertainty, eliminating the need for explicit ensembling or Monte Carlo sampling.

## 3.4 SPIKE-RATE READOUT

Since each LIF neuron (Gerstner & Kistler, 2002) produces binary spikes, we extract rate codes by averaging across $T$ timesteps

$$\mathbf{H}^{(l)} = \frac{1}{T} \sum_{t=1}^{T} \mathbf{Z}_t^{(l)}, \tag{4}$$

where $\mathbf{Z}_t^{(l)} \in \{0,1\}^{N \times d_l}$ is the spike matrix at timestep $t$. These rate codes provide stable representations for downstream prediction and learning.

## 3.5 LOCAL HEBBIAN-STYLE WEIGHT UPDATES

Instead of backpropagation, SIGHT employs local Hebbian updates driven by predictive coding errors. At the top layer $L$, the prediction error is defined as

$$\mathbf{R}^{(L)} = \mathbf{Y} - \mathbf{H}^{(L)}, \tag{5}$$

where $\mathbf{Y}$ is the one-hot label matrix.

For each layer $l$, weights are updated as $\Delta \mathbf{W}^{(l)} \propto (\mathbf{H}^{(l-1)})^\top \mathbf{R}^{(l)}$. This rule couples presynaptic rates $\mathbf{H}^{(l-1)}$ with postsynaptic residuals $\mathbf{R}^{(l)}$, implementing a biologically plausible outer-product Hebbian update. Crucially, no backpropagated error is required and each layer learns from its own residuals.

## 3.6 THEORETICAL ANALYSIS

Here we provide an OOD generalisation bound theorem linking predictive coding error with the generalisation gap, and a convergence theorem showing that iterative error correction in SIGHT reduces an energy function and improves calibration.

**Theorem 1** (OOD generalisation Bound for SIGHT). *Let $\mathcal{H}$ be a hypothesis class of spiking graph neural networks, and let $P_{\mathrm{ID}}$ and $P_{\mathrm{OOD}}$ denote the in-distribution and out-of-distribution data distributions over graphs. For a hypothesis $h \in \mathcal{H}$, the classification error on $P_{\mathrm{OOD}}$ satisfies:*

$$\epsilon_{\mathrm{OOD}}(h) \leq \epsilon_{\mathrm{ID}}(h) + \mathrm{disc}_{\mathcal{H}}(P_{\mathrm{ID}}, P_{\mathrm{OOD}}) + \lambda, \tag{6}$$

*where $\mathrm{disc}_{\mathcal{H}}$ is an $\mathcal{H}$-divergence measuring distribution shift, and $\lambda$ is the minimal joint error over $\mathcal{H}$.*

*Furthermore, in SIGHT, the $\mathcal{H}$-divergence can be upper bounded by a function of the expected predictive coding error $\varepsilon$:*

$$\mathrm{disc}_{\mathcal{H}}(P_{\mathrm{ID}}, P_{\mathrm{OOD}}) \leq C \cdot \mathbb{E}_{G \sim P_{\mathrm{OOD}}}\left[\|\varepsilon(G)\|_2\right], \tag{7}$$

*where $C > 0$ is a constant depending on the Lipschitz smoothness of the message-passing operator. This implies that reducing the predictive coding error $\varepsilon$ directly lowers the bound on the OOD generalisation gap.*

*Proof sketch.* The first inequality follows from the standard domain adaptation bound based on $\mathcal{H}$-divergence. For SIGHT, the predictive coding mechanism produces an error signal $\varepsilon$ at each inference step, which reflects the mismatch between top-down predictions and bottom-up inputs. Since the message-passing operator in SIGHT is $L$-Lipschitz with respect to graph signals, differences in graph distributions translate linearly into differences in predictive coding errors. Bounding the $\mathcal{H}$-divergence in terms of $\mathbb{E}[\|\varepsilon\|]$ follows from this Lipschitz property. $\qquad\square$

**Theorem 2** (Energy Convergence and Calibration Improvement). *Consider SIGHT with an energy function:*

$$E_t = \frac{1}{2} \sum_l \left\| \mathbf{x}_t^l - f^l(\mathbf{x}_t^{l-1}) \right\|_2^2, \tag{8}$$

*where $\mathbf{x}_t^l$ is the latent representation at layer $l$ and time $t$, and $f^l$ is the feedforward mapping at layer $l$.*

*Under the update rule:*

$$\mathbf{x}_{t+1}^l = \mathbf{x}_t^l - \eta \cdot \varepsilon_t^l, \tag{9}$$

*with learning rate $0 < \eta < \frac{2}{L}$, the sequence $\{E_t\}$ is monotonically non-increasing and converges to a fixed point $E^* \geq 0$.*

*Moreover, if the model outputs predictive distributions $p_\theta(y|x)$ with entropy $H_t$ at time $t$, then:*

$$\text{ECE} \leq K \cdot \sqrt{E^*}, \tag{10}$$

*for some constant $K > 0$ depending on the calibration mapping. Thus, reducing the predictive coding error also reduces the Expected Calibration Error (ECE), improving the reliability of uncertainty estimates.*

*Proof sketch.* The energy descent follows from the gradient-descent-like nature of the predictive coding updates, where each latent state moves opposite to the local prediction error. Convergence is ensured by the choice of $\eta$ given the Lipschitz constant of the network layers. The ECE bound follows from relating prediction confidence misalignment to the residual prediction error in the final equilibrium state. $\qquad\square$

Overall, these results provide strong theoretical evidence that predictive coding in SIGHT serves as a unified principle for improving both OOD generalisation and probabilistic calibration. Further details on the proof of convergence for predictive coding inference are provided in Appendix C.

## 4 EXPERIMENTS

We conduct extensive experiments to evaluate the SIGHT framework on five benchmark datasets. Our evaluation considers both node classification accuracy and the reliability of uncertainty estimation under distribution shifts. Additional experimental results are provided in Appendix D.5.

### 4.1 EXPERIMENTAL SETUP

**Datasets.** In this paper, the experiments are conducted on five node classification datasets, i.e., Cora, Citeseer, Pubmed, Twitch, and CBAS. We evaluate SIGHT under two types of distribution shifts: covariate shift and concept shift. For citation networks (i.e., Cora, Citeseer, and PubMed), we apply covariate shift by following a commonly used benchmark (Wu et al., 2022). Specifically, the original node labels retain while synthetically generating spurious node features to induce distribution shifts between the ID and OOD datasets, i.e., $P^{\text{train}}(X) \neq P^{\text{test}}(X)$ but $P^{\text{train}}(Y|X) = P^{\text{test}}(Y|X)$. In contrast, for Twitch and CBAS (Gui et al., 2022), we apply concept shift by altering the conditional distribution while keeping the input distribution stable, i.e., $P^{\text{train}}(Y|X) \neq P^{\text{test}}(Y|X)$. Detailed introduction on datasets and shift types are provided in Appendix D.1.

Table 1: Node classification accuracy and uncertainty calibration on ID and OOD datasets with the GCN backbone. Best results are in **bold**, and SIGHT is marked with ▢. Results with the GAT backbone are in Table 6.

| Method | Accuracy ↑ | | ECE ↓ | | NLL ↓ | | BS ↓ | | AUROC ↑ | |
|---|---|---|---|---|---|---|---|---|---|---|
| | ID | OOD | ID | OOD | ID | OOD | ID | OOD | ID | OOD |
| **Cora** | | | | | | | | | | |
| GCN | 89.8±0.3 | 43.5±5.2 | 0.019±0.003 | 0.391±0.071 | 0.318±0.008 | 2.595±0.512 | 0.150±0.004 | 0.919±0.103 | **88.7±0.3** | 59.9±3.7 |
| +G-ΔUQ | 91.3±0.4 | 80.1±4.7 | 0.019±0.004 | 0.053±0.031 | 0.270±0.018 | 0.574±0.149 | 0.130±0.007 | 0.288±0.068 | 87.3±1.2 | 81.5±2.8 |
| +SIGHT | **93.2±0.6** | **95.6±0.2** | **0.015±0.002** | **0.009±0.002** | **0.241±0.021** | **0.153±0.010** | **0.105±0.009** | **0.068±0.003** | 86.7±1.2 | **89.8±1.5** |
| **Citeseer** | | | | | | | | | | |
| GCN | 82.1±0.4 | 46.6±1.9 | 0.026±0.003 | 0.334±0.027 | 0.503±0.012 | 1.838±0.051 | 0.252±0.006 | 0.793±0.026 | **84.1±0.8** | 74.1±1.6 |
| +G-ΔUQ | 81.6±0.4 | 71.0±4.2 | **0.021±0.008** | 0.066±0.027 | 0.531±0.037 | 0.889±0.145 | 0.263±0.010 | 0.416±0.056 | 82.2±1.6 | 76.5±1.3 |
| +SIGHT | **86.6±0.7** | **91.2±0.4** | 0.028±0.006 | **0.024±0.004** | **0.452±0.023** | **0.307±0.016** | **0.204±0.010** | **0.136±0.008** | 80.1±1.1 | **83.4±0.8** |
| **Pubmed** | | | | | | | | | | |
| GCN | 88.0±0.1 | 73.7±1.9 | 0.008±0.001 | 0.146±0.028 | 0.306±0.001 | 0.960±0.081 | 0.173±0.000 | 0.411±0.028 | 85.2±0.3 | 72.3±1.5 |
| +G-ΔUQ | 91.3±0.2 | 83.8±0.7 | **0.006±0.001** | 0.090±0.009 | **0.236±0.002** | 0.668±0.048 | **0.129±0.002** | 0.263±0.013 | **86.3±0.4** | 74.2±0.3 |
| +SIGHT | 90.0±0.1 | **90.1±0.2** | 0.009±0.001 | **0.006±0.001** | 0.275±0.001 | **0.275±0.004** | 0.149±0.001 | **0.148±0.002** | 85.5±0.4 | **85.1±0.2** |
| **Twitch** | | | | | | | | | | |
| GCN | 60.7±9.0 | 57.5±4.1 | 0.129±0.066 | 0.046±0.018 | 0.685±0.006 | 0.683±0.007 | 0.492±0.006 | 0.490±0.008 | **59.5±4.3** | 49.2±3.3 |
| +G-ΔUQ | **64.8±9.9** | 58.3±2.9 | 0.163±0.067 | 0.065±0.024 | 0.690±0.021 | 0.687±0.009 | 0.497±0.021 | 0.494±0.009 | 58.0±9.1 | 45.4±1.5 |
| +SIGHT | 59.3±6.6 | **60.6±1.4** | **0.077±0.033** | **0.039±0.009** | **0.676±0.035** | **0.671±0.005** | **0.483±0.033** | **0.478±0.005** | 56.1±6.9 | **52.2±3.9** |
| **CBAS** | | | | | | | | | | |
| GCN | 77.0±4.9 | 69.6±4.9 | 0.095±0.025 | 0.116±0.007 | 0.596±0.032 | 0.840±0.026 | 0.327±0.025 | 0.432±0.022 | 77.5±5.0 | 71.6±2.6 |
| +G-ΔUQ | 76.4±5.2 | 65.1±2.8 | 0.125±0.018 | 0.131±0.039 | 0.611±0.058 | 0.881±0.043 | 0.337±0.036 | 0.461±0.031 | 77.4±6.4 | 66.9±7.4 |
| +SIGHT | **93.9±2.2** | **76.4±2.8** | **0.071±0.021** | **0.086±0.020** | **0.241±0.067** | **0.746±0.075** | **0.114±0.034** | **0.363±0.032** | **81.6±4.9** | **73.0±1.8** |

**Baselines.** We adopt GCN (Kipf & Welling, 2017) and GAT (Veličković et al., 2018) as backbones since they form the foundational frameworks of most GNN models and are widely recognizee as important and representative architectures in graph learning. In addition, we consider G-ΔUQ (Trivedi et al., 2024), a recently proposed framework for uncertainty quantification on graphs. G-ΔUQ introduces graph-specific anchoring strategies that regularise the predictive distribution, thereby improving intrinsic uncertainty estimates. Unlike standard post-hoc calibration, it integrates uncertainty modelling directly into the training objective, making it a strong representative of training-based approaches. Finally, we also demonstrate that the combination of our framework with different post-hoc calibration models can further enhance calibration (see Appendix D.2).

**Evaluation Metrics.** We evaluate performance using accuracy alongside four complementary calibration-oriented metrics: expected calibration error (ECE) (Guo et al., 2017), negative log-likelihood (NLL), Brier Score (BS), and area under the receiver operating characteristic curve (AUROC). Details of the metrics are in Appendix D.3.

**Implementation.** SIGHT is implemented with three layers of spiking graph convolution, each followed by predictive coding inference with $K = 20$ iterations. Input node features are encoded into Poisson spike trains with $T = 25$ time steps, and hidden dimensions are set to 128–128–64. For Cora, Citeseer, and PubMed, we use 500 training epochs with early stopping (patience = 100); for CBAS and Twitch, we use 1000 epochs with patience = 200. The learning rates are set to $\eta_x = 0.005$ and $\eta_p = 0.0005$ for Cora, Citeseer, PubMed, and CBAS, while higher rates $\eta_x = 0.01$ and $\eta_p = 0.001$ are used for Twitch. Each experiment is repeated with five random seeds for statistical robustness. The full implementation details can be found in Appendix D.4.

## 4.2 RESULTS

**Comparison of Performance and Calibration.** Table 1 compares the node classification accuracy and calibration capabilities of the traditional GNN model on ID and OOD data after introducing different intrinsic calibration methods: G-ΔUQ and SIGHT. Overall, SIGHT achieves the best performance on the majority of benchmarks, particularly in OOD regimes. For example, on Cora and Citeseer, SIGHT significantly improves OOD accuracy by even 40% while also reducing ECE by 40 times. On Pubmed, SIGHT attains balanced gains, achieving strong ID accuracy and the best results on all OOD scenarios, indicating superior uncertainty awareness. On Twitch and CBAS, SIGHT consistently achieves lower calibration errors together with higher accuracy and AUROC, confirming its strong generalization and uncertainty estimation capabilities under different OOD conditions. These results highlight that *SIGHT not only matches or surpasses accuracy-oriented baselines but also establishes new state-of-the-art calibration and OOD generalisation performance.*

When combining SIGHT with different post-hoc calibration models, as shown in Table 2, SIGHT consistently produces better-calibrated models, with Accuracy and ECE values outperforming those

Table 2: Accuracy and ECE under distribution shifts for different post-hoc calibration models using GCN and GAT backbones, w/o and w/ SIGHT ( ). ✗ denotes no post-hoc calibration.

| | Model | GCN Accuracy ↑ w/o SIGHT | w/ SIGHT | ECE ↓ w/o SIGHT | w/ SIGHT | GAT Accuracy ↑ w/o SIGHT | w/ SIGHT | ECE ↓ w/o SIGHT | w/ SIGHT |
|---|---|---|---|---|---|---|---|---|---|
| **Cora** | ✗ | 43.5±5.2 | 95.6±0.2 | 0.391±0.071 | 0.009±0.002 | 67.4±4.1 | 96.3±0.7 | 0.164±0.026 | 0.008±0.002 |
| | CAGCN | 40.9±3.3 | 95.6±0.2 | 0.439±0.081 | 0.037±0.019 | 67.4±4.1 | 95.8±0.7 | 0.255±0.031 | 0.042±0.022 |
| | Dirichlet | 42.1±4.4 | 95.5±0.1 | 0.412±0.039 | 0.012±0.001 | 65.9±3.0 | 96.6±0.4 | 0.174±0.024 | 0.009±0.003 |
| | ETS | 42.1±5.7 | 95.6±0.2 | 0.419±0.055 | 0.009±0.001 | 67.0±3.9 | 95.2±1.0 | 0.176±0.017 | 0.011±0.003 |
| | GATS | 42.5±6.2 | 95.4±0.3 | 0.433±0.085 | 0.051±0.020 | 66.9±4.2 | 96.5±0.5 | 0.262±0.042 | 0.061±0.039 |
| | IRM | 41.8±4.4 | 95.4±0.2 | 0.434±0.045 | 0.009±0.001 | 67.4±4.1 | 96.6±0.3 | 0.188±0.021 | 0.008±0.001 |
| | Order | 41.1±3.9 | 95.5±0.1 | 0.407±0.031 | 0.012±0.005 | 66.6±3.8 | 96.4±0.3 | 0.179±0.025 | 0.009±0.002 |
| | Spline | 41.4±4.1 | 95.5±0.2 | 0.427±0.043 | 0.286±0.003 | 66.1±2.7 | 96.1±0.8 | 0.178±0.022 | 0.287±0.016 |
| | VS | 42.7±5.8 | 95.4±0.3 | 0.419±0.040 | 0.007±0.001 | 67.0±3.9 | 96.8±0.5 | 0.176±0.027 | 0.007±0.003 |
| **Citeseer** | ✗ | 46.6±1.9 | 91.2±0.4 | 0.334±0.027 | 0.024±0.004 | 46.6±1.9 | 93.4±0.2 | 0.334±0.027 | 0.028±0.003 |
| | CAGCN | 46.9±2.1 | 90.9±0.2 | 0.448±0.057 | 0.036±0.008 | 66.5±3.2 | 92.7±0.2 | 0.257±0.027 | 0.041±0.008 |
| | Dirichlet | 47.1±2.0 | 90.9±0.3 | 0.333±0.028 | 0.025±0.002 | 66.3±3.1 | 92.8±0.5 | 0.123±0.023 | 0.029±0.006 |
| | ETS | 46.7±1.9 | 90.6±0.4 | 0.335±0.026 | 0.025±0.004 | 66.4±3.3 | 93.1±0.5 | 0.132±0.025 | 0.036±0.004 |
| | GATS | 47.0±2.1 | 90.9±0.2 | 0.459±0.050 | 0.072±0.011 | 67.1±3.4 | 93.3±0.3 | 0.293±0.031 | 0.052±0.003 |
| | IRM | 46.7±1.9 | 90.7±0.2 | 0.330±0.029 | 0.027±0.003 | 66.6±3.2 | 92.7±0.6 | 0.126±0.027 | 0.024±0.002 |
| | Order | 47.2±2.5 | 90.9±0.2 | 0.317±0.024 | 0.024±0.004 | 67.2±3.3 | 93.1±0.1 | 0.126±0.014 | 0.027±0.003 |
| | Spline | 46.8±2.0 | 90.9±0.1 | 0.315±0.026 | 0.173±0.002 | 66.3±3.0 | 92.7±0.7 | 0.125±0.029 | 0.177±0.002 |
| | VS | 46.9±2.1 | 90.7±0.4 | 0.334±0.035 | 0.021±0.002 | 66.2±2.9 | 92.8±0.1 | 0.125±0.020 | 0.024±0.003 |
| **Pubmed** | ✗ | 73.7±1.9 | 90.1±0.2 | 0.146±0.028 | 0.006±0.001 | 73.7±1.9 | 85.7±0.3 | 0.146±0.028 | 0.011±0.002 |
| | CAGCN | 74.8±1.9 | 90.1±0.2 | 0.074±0.017 | 0.037±0.013 | 77.4±2.5 | 85.6±0.2 | 0.113±0.043 | 0.052±0.022 |
| | Dirichlet | 74.2±1.6 | 90.3±0.2 | 0.134±0.024 | 0.009±0.001 | 77.0±1.8 | 85.6±0.1 | 0.106±0.016 | 0.017±0.003 |
| | ETS | 74.9±1.9 | 90.2±0.2 | 0.125±0.029 | 0.007±0.001 | 77.2±2.3 | 85.8±0.4 | 0.104±0.021 | 0.010±0.003 |
| | GATS | 73.8±1.7 | 90.2±0.1 | 0.118±0.033 | 0.059±0.019 | 77.3±2.4 | 85.9±0.2 | 0.129±0.025 | 0.079±0.025 |
| | IRM | 74.2±1.5 | 90.2±0.2 | 0.141±0.021 | 0.005±0.001 | 77.2±1.9 | 85.5±0.4 | 0.114±0.022 | 0.007±0.002 |
| | Order | 74.1±1.8 | 90.3±0.2 | 0.136±0.022 | 0.007±0.001 | 77.1±2.0 | 86.1±0.2 | 0.114±0.025 | 0.009±0.001 |
| | Spline | 73.8±1.5 | 90.2±0.1 | 0.139±0.021 | 0.084±0.001 | 76.7±1.6 | 85.9±0.3 | 0.116±0.019 | 0.085±0.009 |
| | VS | 73.7±1.2 | 90.3±0.1 | 0.148±0.016 | 0.006±0.000 | 77.0±1.9 | 86.4±0.5 | 0.112±0.019 | 0.009±0.001 |
| **Twitch** | ✗ | 57.5±4.1 | 60.6±1.4 | 0.046±0.028 | 0.039±0.009 | 57.5±4.1 | 60.1±3.3 | 0.046±0.028 | 0.031±0.027 |
| | CAGCN | 54.4±4.2 | 60.4±1.3 | 0.079±0.004 | 0.091±0.012 | 56.0±5.2 | 61.3±2.2 | 0.072±0.015 | 0.066±0.009 |
| | Dirichlet | 55.7±3.9 | 60.4±1.3 | 0.016±0.006 | 0.012±0.002 | 55.9±5.3 | 60.4±3.0 | 0.031±0.017 | 0.019±0.009 |
| | ETS | 56.3±3.3 | 59.8±1.6 | 0.037±0.031 | 0.030±0.010 | 55.6±5.6 | 60.2±2.4 | 0.063±0.062 | 0.023±0.011 |
| | GATS | 55.2±3.8 | 60.4±1.6 | 0.050±0.020 | 0.091±0.018 | 55.8±5.4 | 61.2±2.3 | 0.035±0.018 | 0.058±0.011 |
| | IRM | 55.6±3.5 | 60.4±1.7 | 0.009±0.001 | 0.011±0.001 | 55.5±5.7 | 61.4±1.1 | 0.010±0.001 | 0.012±0.003 |
| | Order | 54.1±4.4 | 60.6±1.5 | 0.024±0.025 | 0.018±0.007 | 55.3±6.0 | 60.6±3.5 | 0.036±0.011 | 0.019±0.008 |
| | Spline | 54.8±4.1 | 60.2±1.4 | 0.140±0.029 | 0.026±0.015 | 55.6±5.6 | 61.8±1.1 | 0.099±0.106 | 0.034±0.024 |
| | VS | 55.5±3.8 | 60.8±1.4 | 0.040±0.026 | 0.011±0.002 | 55.8±5.4 | 60.7±2.1 | 0.039±0.011 | 0.015±0.004 |
| **CBAS** | ✗ | 69.6±4.9 | 76.4±2.8 | 0.116±0.007 | 0.086±0.020 | 70.4±2.7 | 59.9±3.0 | 0.112±0.022 | 0.114±0.039 |
| | CAGCN | 65.4±3.6 | 72.2±1.9 | 0.115±0.036 | 0.124±0.022 | 66.0±7.4 | 57.8±2.6 | 0.125±0.038 | 0.165±0.017 |
| | Dirichlet | 67.8±3.0 | 76.7±1.8 | 0.090±0.013 | 0.092±0.019 | 66.0±7.4 | 61.8±2.4 | 0.085±0.017 | 0.116±0.024 |
| | ETS | 68.5±4.1 | 76.7±2.1 | 0.087±0.026 | 0.080±0.012 | 65.8±7.2 | 59.3±2.6 | 0.130±0.055 | 0.139±0.037 |
| | GATS | 68.4±4.4 | 78.5±1.3 | 0.149±0.090 | 0.135±0.029 | 65.7±7.1 | 62.3±3.3 | 0.131±0.039 | 0.129±0.025 |
| | IRM | 68.7±4.3 | 76.7±2.8 | 0.088±0.009 | 0.102±0.020 | 63.1±6.3 | 60.5±1.3 | 0.099±0.039 | 0.130±0.027 |
| | Order | 66.3±5.1 | 77.1±1.8 | 0.069±0.020 | 0.098±0.016 | 65.2±6.8 | 59.3±2.5 | 0.118±0.053 | 0.129±0.020 |
| | Spline | 68.7±4.4 | 76.7±2.8 | 0.136±0.099 | 0.126±0.015 | 67.1±8.8 | 58.7±2.7 | 0.152±0.058 | 0.143±0.030 |
| | VS | 67.5±3.3 | 74.8±2.6 | 0.101±0.029 | 0.097±0.009 | 69.2±7.4 | 58.4±1.7 | 0.095±0.011 | 0.094±0.024 |

of the vanilla (w/o SIGHT) models in all datasets. For Cora and Citeseer, incorporating SIGHT into GNNs can increase even more than 50% Accuracy and ECE can be reduced even to 60 times that without SIGHT. Furthermore, integrating SIGHT with post-hoc calibration models enhances performance more effectively than applying the same calibration strategy to vanilla models. For a given post-hoc method, SIGHT not only improves calibration but also maintains or even surpasses the accuracy of baseline models across most datasets. Although certain combinations of SIGHT and post-hoc calibration methods may lead to slight performance degradation (e.g., SIGHT with GAT on CBAS), this should not be viewed as a limitation of SIGHT itself. Other post-hoc methods applied in the same setting also fail to yield significant improvements in either accuracy or calibration, and in some cases even cause further degradation. In contrast, when combined with GCN, SIGHT already achieves the best performance on CBAS. This suggests that the benefits of SIGHT are more effectively realised through appropriate backbone selection rather than additional post-hoc calibration, highlighting its inherent ability to deliver robust generalisation and well-calibrated uncertainty estimates. Overall, *SIGHT consistently yields better-calibrated models for node classification tasks and can be combined with post-hoc calibration models for further gains.*

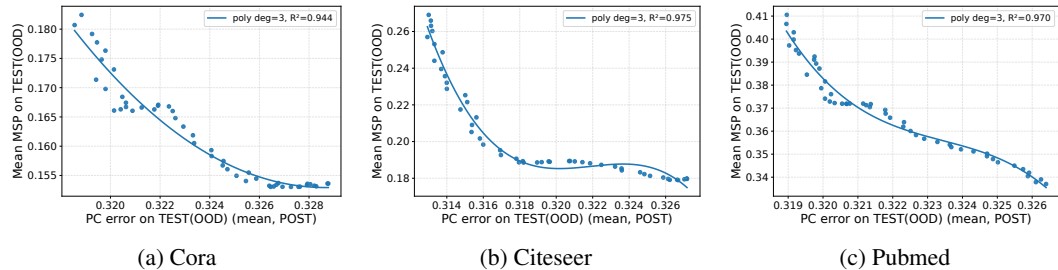

Figure 2: Correlation between predictive coding (PC) error and confidence scores on the Cora, Citeseer, and Pubmed OOD test sets. Higher PC errors consistently correspond to lower confidence, showing that PC residuals serve as an intrinsic measure of uncertainty.

**OOD Detection.** OOD detection (Hendrycks et al., 2019; Ren et al., 2023) aims to classify samples as ID or OOD. To evaluate whether models assign higher uncertainty to shifted inputs (Bazhenov et al., 2023), we compare G-$\Delta$UQ and SIGHT with GCN and GAT backbones. Their AUROC results are reported in Table 3.

Across the three citation networks (Cora, Citeseer, and Pubmed) and the social network Twitch, SIGHT consistently achieves the highest AUROC under GCN backbone, substantially outperforming plain GCN and even the strong G-$\Delta$UQ baseline nearly 20% and 7% at most, respectively. For the CBAS dataset, SIGHT with GAT remains competitive and surpasses the baselines, confirming its robustness under diverse distribution shifts. These improvements highlight the effectiveness of predictive coding residuals in distinguishing ID from OOD samples. Overall, these results demonstrate that *SIGHT provides reliable uncertainty estimation for OOD detection, outperforming both conventional GNNs and advanced calibration baselines.*

Table 3: AUROC results for OOD detection. Best results are in **bold**, and SIGHT is marked with ▢.

| Method | Cora | Citeseer | Pubmed | Twitch | CBAS |
|--------|------|----------|--------|--------|------|
| GCN | 70.4±2.8 | 74.1±1.1 | 72.6±1.2 | 53.2±0.5 | 74.5±3.8 |
| +G-$\Delta$UQ | 83.0±2.2 | 77.7±1.0 | 74.8±0.4 | 52.2±4.6 | 72.3±6.6 |
| +SIGHT | **89.0±1.6** | **84.0±1.2** | **85.1±0.2** | **54.5±2.4** | 78.1±2.4 |
| GAT | 73.9±2.1 | 78.3±1.1 | 77.0±0.6 | 49.5±2.4 | 75.9±4.1 |
| +G-$\Delta$UQ | 83.3±2.4 | 80.4±1.1 | 82.7±0.8 | 49.3±3.9 | 78.3±5.3 |
| +SIGHT | 88.0±2.8 | 81.1±1.8 | 82.1±0.2 | 52.7±3.1 | **83.1±3.4** |

**Explainability of Predictive Coding on Uncertainty Estimation.** This experiment measures PC error as an interpretability signal in our model trained on OOD datasets. Figure 2 illustrates the correlation between PC error and model confidence measured by the mean maximum softmax probability (MSP), on the OOD splits of Cora, Citeseer, and Pubmed. Across all three datasets, we observe a strong negative correlation: as the average PC error increases, the MSP decreases. This indicates that larger residuals correspond to lower confidence, thereby providing an intrinsic and interpretable signal of uncertainty. Notably, the fitted polynomial curves show high $R^2$ values ($> 0.94$), confirming that PC errors serve as a reliable proxy for confidence across diverse graph benchmarks. These results highlight that *SGPC does not require additional calibration procedures to extract meaningful uncertainty estimates, as the residuals themselves naturally align with confidence scores under distribution shifts.* See Appendix D.5.2 for more experimental results on Twitch and CBAS datasets.

**Ablation Study.** Table 4 reports an ablation study with GCN and GAT backbone, where we compare the full SIGHT model against three variants: without spiking neurons (w/o Spik), without predictive coding (w/o PC), and without the joint Spiking Predictive Coding mechanism (w/o SPC). The complete SIGHT model consistently achieves the best performance across all datasets and metrics, confirming the effectiveness of combining spiking computation with predictive coding. When the spiking mechanism is removed (w/o Spiking), accuracy drops substantially (e.g., from 95.6% to 68.4% on Cora with GCN) and calibration metrics such as ECE and Brier Score deteriorate, indicating that event-driven representations are key to both generalisation and calibration. Excluding predictive coding (w/o PC) yields relatively high accuracy on some datasets (e.g., Pubmed with GAT, 92.9%), but calibration performance significantly worsens, as reflected by higher NLL and ECE values, showing that local error feedback is essential for uncertainty estimation and well-calibrated

Table 4: Ablation study of SIGHT with GCN and GAT backbones. Best results are in **bold**.

| | Method | GCN | | | | | GAT | | | | |
|---|---|---|---|---|---|---|---|---|---|---|---|
| | | Acc ↑ | ECE ↓ | NLL ↓ | BS ↓ | AUROC ↑ | Acc ↑ | ECE ↓ | NLL ↓ | BS ↓ | AUROC ↑ |
| Cora | SIGHT | **95.6±0.2** | **0.009±0.002** | **0.153±0.010** | **0.068±0.003** | **89.8±1.5** | 96.3±0.7 | **0.008±0.002** | **0.138±0.028** | 0.058±0.012 | **89.6±2.1** |
| | w/o Spik | 68.4±5.1 | 0.146±0.043 | 1.011±0.196 | 0.473±0.084 | 71.4±2.3 | 64.5±5.5 | 0.175±0.048 | 1.243±0.230 | 0.551±0.093 | 66.3±4.7 |
| | w/o PC | 95.2±0.3 | 0.020±0.005 | 0.158±0.004 | 0.076±0.002 | 89.3±1.4 | **97.1±0.2** | 0.017±0.005 | 0.135±0.022 | **0.048±0.006** | 83.3±5.2 |
| | w/o SPC | 43.5±5.2 | 0.391±0.071 | 2.595±0.512 | 0.919±0.103 | 59.9±3.7 | 67.4±4.1 | 0.164±0.026 | 1.072±0.108 | 0.498±0.059 | 71.1±3.2 |
| Citeseer | SIGHT | 91.2±0.4 | **0.024±0.004** | 0.307±0.016 | 0.136±0.008 | **83.4±0.8** | **93.4±0.2** | **0.028±0.003** | **0.261±0.012** | **0.105±0.005** | **80.8±1.2** |
| | w/o Spik | 75.0±2.7 | 0.040±0.007 | 0.793±0.081 | 0.372±0.036 | 75.6±1.4 | 79.1±3.6 | 0.038±0.009 | 0.678±0.096 | 0.315±0.047 | 78.3±1.3 |
| | w/o PC | **92.1±0.2** | 0.037±0.005 | **0.296±0.011** | **0.126±0.004** | 82.3±1.3 | 93.2±0.3 | 0.051±0.007 | 0.287±0.009 | 0.110±0.005 | 79.0±1.9 |
| | w/o SPC | 46.6±1.9 | 0.334±0.027 | 1.838±0.051 | 0.793±0.026 | 74.1±1.6 | 66.6±3.3 | 0.125±0.022 | 0.968±0.085 | 0.470±0.037 | 77.4±1.3 |
| Pubmed | SIGHT | **90.1±0.2** | **0.006±0.001** | **0.275±0.004** | **0.148±0.002** | **85.1±0.2** | 85.7±0.3 | 0.011±0.002 | 0.380±0.009 | 0.210±0.005 | 81.7±0.8 |
| | w/o Spik | 78.3±1.2 | 0.116±0.013 | 0.665±0.043 | 0.338±0.018 | 77.1±1.6 | 79.1±3.6 | 0.038±0.009 | 0.678±0.096 | 0.315±0.047 | 78.3±1.3 |
| | w/o PC | 87.2±0.1 | 0.008±0.001 | 0.345±0.002 | 0.189±0.001 | 81.8±0.1 | **92.9±0.2** | **0.008±0.001** | **0.203±0.006** | **0.107±0.003** | **87.3±0.4** |
| | w/o SPC | 73.7±1.9 | 0.146±0.028 | 0.960±0.081 | 0.411±0.028 | 72.3±1.5 | 76.7±1.5 | 0.116±0.018 | 0.776±0.078 | 0.352±0.022 | 76.0±0.7 |
| Twitch | SIGHT | **60.6±1.4** | 0.039±0.009 | **0.671±0.005** | 0.478±0.005 | **52.2±3.9** | **60.1±3.3** | 0.031±0.027 | **0.670±0.013** | **0.477±0.012** | **51.6±4.7** |
| | w/o Spik | 59.5±5.5 | 0.016±0.003 | 0.669±0.012 | **0.476±0.012** | 50.2±4.5 | 55.4±7.9 | 0.072±0.041 | 0.681±0.014 | 0.488±0.014 | 46.6±7.6 |
| | w/o PC | 59.6±2.2 | **0.035±0.021** | 0.675±0.011 | 0.482±0.010 | 51.8±4.1 | 59.6±2.9 | 0.038±0.029 | 0.673±0.012 | 0.481±0.012 | 51.3±5.1 |
| | w/o SPC | 57.5±4.1 | 0.046±0.028 | 0.683±0.007 | 0.490±0.008 | 49.2±3.3 | 58.6±3.7 | 0.036±0.017 | 0.678±0.012 | 0.485±0.012 | 49.7±2.6 |
| CBAS | SIGHT | **76.4±2.8** | **0.086±0.020** | 0.746±0.075 | 0.363±0.032 | 73.0±1.8 | 60.0±3.0 | 0.114±0.039 | 0.978±0.054 | 0.504±0.029 | **81.0±5.3** |
| | w/o Spik | 74.8±6.2 | 0.093±0.021 | 0.726±0.105 | 0.359±0.066 | 73.8±3.5 | **73.0±2.8** | **0.088±0.019** | **0.802±0.030** | **0.396±0.008** | 72.3±6.7 |
| | w/o PC | 76.0±3.8 | 0.097±0.009 | **0.645±0.115** | **0.334±0.061** | **82.8±7.3** | 70.4±3.1 | 0.122±0.033 | 0.839±0.046 | 0.427±0.025 | 76.0±4.9 |
| | w/o SPC | 69.6±4.9 | 0.116±0.007 | 0.840±0.026 | 0.432±0.022 | 71.6±2.6 | 70.4±2.6 | 0.112±0.022 | 0.836±0.074 | 0.429±0.039 | 70.4±2.8 |

predictions. Finally, removing the full spiking predictive coding loop (w/o SPC) leads to the most severe degradation across metrics. These results collectively demonstrate that *both spiking computation and predictive coding contribute complementary strengths. Their integration in SIGHT is thus crucial for delivering high accuracy, strong calibration, and reliable OOD generalization.*

From the perspective of dataset characteristics, it can also be observed that incorporating SIGHT into the same backbone consistently yields strong performance across different types of distribution shifts. Specifically, integrating SIGHT with GCN leads to better results on Cora, Pubmed, and CBAS, while combining it with GAT achieves the best performance on Citeseer and Twitch. This divergence can be attributed to structural differences between datasets. These findings suggest that *the choice of backbone architecture should be tailored to dataset properties*, and highlight the flexibility of SIGHT in enhancing a wide range of GNN models under both feature and concept shifts.

**Parameter Analysis.** We further investigate the impact of different hyperparameters on SIGHT, including the number of predictive coding iterations $K$, the number of timesteps $T$, and the learning rate under both GCN and GAT backbones. The results show that SIGHT is generally robust to variations in $K$ and $T$, while the learning rate has a stronger influence, particularly on smaller datasets. These findings confirm the practicality and robustness of SIGHT, with detailed experimental results provided in Appendix D.5.3.

## 5 CONCLUSION

In this work, we introduced SpIking GrapH predicTive coding (SIGHT), a novel brain-inspired framework that integrates spiking neural dynamics with predictive coding to achieve uncertainty-aware graph learning. SIGHT departs from deterministic GNNs by iteratively minimizing prediction errors across spiking graph states, enabling both robust generalisation under distribution shifts and dynamic confidence estimation during inference. Experimental results demonstrate that SIGHT provides robust uncertainty estimation under distribution shifts while maintaining competitive predictive performance, paving the way toward more reliable and neuromorphic-ready graph learning models. Looking forward, we aim to extend SIGHT to multimodal graph learning, incorporating heterogeneous sensory or semantic inputs, and explore its deployment on neuromorphic hardware platforms, leveraging the event-driven and energy-efficient properties of spiking computation for real-time, low-power applications. In addition, future work will investigate how replacing traditional backpropagation with predictive coding in deep neural networks influences model calibration under out-of-distribution scenarios.

**Reproducibility Statement.** We are committed to ensuring reproducibility of our work. Details of model architectures, training procedures, and evaluation protocols are provided in Section 4.1 and Appendix D, including hyperparameters, training epochs, and optimisation strategies. All datasets (Cora, Citeseer, Pubmed, Twitch, and CBAS) are publicly available with standard preprocessing following prior works. To further support reproducibility, we release open-source code containing implementations of SIGHT, baselines, and post-hoc calibration models, together with configuration files for reproducing all experiments and ablation studies. Pseudocode and the anonymous download link are given in Appendix B.

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

# A  THE USE OF LARGE LANGUAGE MODELS (LLMS)

In this paper, LLMs were used as a general-purpose assist tool during the preparation of this paper. Specifically, LLMs assisted with polishing grammar of the manuscript and rephrasing sentences for conciseness and readability.

# B  METHODOLOGY

## B.1  REPRODUCIBILITY

To facilitate reproducibility, we provide our code in an anonymous repository. Additional details regarding the benchmarks and experimental setup are presented in Appendix D.

## B.2  LEAKY INTEGRATE-AND-FIRE (LIF)

In SIGHT, the Leaky Integrate-and-Fire (LIF) neuron plays a central role by converting continuous prediction errors into discrete spike events and updating latent states through biologically inspired neural dynamics. By acting as a leaky integrator with a firing threshold, the LIF mechanism approximates gradient descent on the predictive coding energy function while preserving event-driven sparsity. This not only ensures biological plausibility but also provides computational efficiency and energy savings, making the model well-suited for deployment on neuromorphic hardware.

**From Predictive Coding to Input Current.**   After $K$ inference steps of the predictive-coding loop in layer $\ell$, we obtain a continuous latent drive $\mathbf{z}_K^{(\ell)}$. This value acts as the constant synaptic current fed into the LIF neurons:

$$\mathbf{i}^{(\ell)} := \mathbf{z}_K^{(\ell)}. \tag{11}$$

**LIF Recurrence and Spiking.**   Given the input $\mathbf{i}^{(\ell)}$, the LIF membrane potential is updated across $T$ spiking timesteps as

$$\mathbf{v}_{t+1}^{(\ell)} = \beta\,\mathbf{v}_t^{(\ell)} + \mathbf{i}^{(\ell)} - v_{\text{th}}\,\mathbf{s}_t^{(\ell)}, \quad t = 0, \dots, T-1, \tag{12}$$

$$\mathbf{s}_t^{(\ell)} = \Theta\!\left(\mathbf{v}_t^{(\ell)} - v_{\text{th}}\right), \tag{13}$$

where $0 < \beta < 1$ is the leak factor ($\beta = 0.9$ in our experiments), $v_{\text{th}}$ is the firing threshold, and $\Theta(\cdot)$ denotes the Heaviside step function. Each $\mathbf{s}_t^{(\ell)} \in \{0,1\}^{N \times d_\ell}$ is a binary spike matrix indicating whether neurons fire at timestep $t$.

**Rate-code Representation.**   To obtain a stable representation from the spike train, we compute the average firing rate across all timesteps:

$$\mathbf{H}^{(\ell)} = \frac{1}{T}\sum_{t=1}^{T} \mathbf{s}_t^{(\ell)}. \tag{14}$$

This rate code $\mathbf{H}^{(\ell)} \in [0,1]^{N \times d_\ell}$ is then passed to the next layer or the final classifier.

Thus, each LIF block performs the following transformation:

$$\mathbf{z}_K^{(\ell)} \;\longrightarrow\; \{\mathbf{s}_t^{(\ell)}\}_{t=1}^{T} \;\longrightarrow\; \mathbf{H}^{(\ell)}, \tag{15}$$

bridging predictive coding states with spiking dynamics.

## B.3  SUMMARY OF SIGHT

Figure 1 illustrates the overall framework of SIGHT. In traditional GNNs, node representations are updated by aggregating neighbor features through weighted message passing and nonlinear transformations, with embeddings refined globally via backpropagation. While this synchronous aggregation captures structural context, it lacks mechanisms for uncertainty modeling and local error

---

**Algorithm 1** Training Epoch of SIGHT

---

**Require:** Node features $\mathbf{X}$, adjacency $\tilde{\mathbf{A}}$, labels $\mathbf{Y}$
 1: **Poisson Encoding:** Generate spike trains $\mathbf{X}_{\text{spikes}}$ from normalized features.
 2: **Predictive Coding Inference:**
 3: **for** each layer $l = 1$ to $L$ **do**
 4:     Input $\mathbf{H}^{(0)} \leftarrow \mathbf{X}_{\text{spikes}}$
 5:     Pre-activation $\mathbf{P}^{(l)} \leftarrow \tilde{\mathbf{A}}\mathbf{H}^{(l-1)}\mathbf{W}^{(l)}$
 6:     Iteratively update latent state $\mathbf{Z}^{(l)}$ via LIF error–prediction dynamics.
 7: **end for**
 8: **Rate Readout:** $\mathbf{H}^{(l)} \leftarrow \frac{1}{T}\sum_t \mathbf{Z}_t^{(l)}$
 9: **Hebbian Learning:**
10: **for** each layer $l = 1$ to $L$ **do**
11:     Update $\mathbf{W}^{(l)} \propto (\mathbf{H}^{(l-1)})^\top(\mathbf{Y} - \mathbf{H}^{(L)})$
12: **end for**

---

correction. In contrast, SIGHT augments aggregation with predictive coding units that exchange both predictive signals and residual errors as spikes. This enables bi-directional message passing, allowing nodes to iteratively refine their states based on mismatches between expected and observed inputs. As a result, embeddings become not only structurally aware but also uncertainty-informed and energy-efficient.

### B.4  PSEUDOCODE

Please refer to Algorithm 1 for the pseudocode.

### B.5  COMPUTATIONAL COMPLEXITY ANALYSIS

We conduct energy efficiency analysis and time complexity analysis.

**Energy Efficiency Analysis.** Spiking implementations introduce event-driven sparsity with only a fraction $\rho \ll 1$ of neurons fire per timestep. The effective cost per timestep is therefore $\mathcal{O}\big(\rho|\mathcal{E}|d_{l-1} + \rho N d_l\big)$. This makes SIGHT well-suited for neuromorphic hardware, where multiplications are replaced by accumulations triggered by spikes, reducing both latency and power consumption.

**Time Complexity.** For each layer $l$ with input dimension $d_{l-1}$ and output dimension $d_l$, a single graph convolution requires $\mathcal{O}\big(|\mathcal{E}|d_{l-1} + N d_l\big)$, where $|\mathcal{E}|$ is the number of edges. Predictive coding inference introduces an additional factor of $K$ inner iterations per layer. Thus, the per-layer cost is $\mathcal{O}\big(K(|\mathcal{E}|d_{l-1} + N d_l)\big)$. Over $L$ layers and $T$ Poisson timesteps, the total complexity is $\mathcal{O}\big(TK\sum_{l=1}^{L}(|\mathcal{E}|d_{l-1} + N d_l)\big)$. In practice, $T$ and $K$ are small constants, so the overhead remains linear in $|\mathcal{E}|$ and $N$.

**Space Complexity.** Unlike backpropagation, SIGHT does not store full computational graphs or intermediate gradients. Each layer only maintains its latent state $\mathbf{Z}^{(l)}$, residuals $\mathbf{E}^{(l)}$, and weight matrix $\mathbf{W}^{(l)}$. Hence, memory scales as $\mathcal{O}\big(\sum_{l=1}^{L} N d_l + d_{l-1}d_l\big)$, significantly lower than $\mathcal{O}(T\sum_l N d_l)$ required by backpropagation through time in spiking networks.

**Comparison with Backpropagation.** Backpropagation requires gradient chaining across all layers and timesteps, incurring $\mathcal{O}(LT)$ memory and repeated backward passes. SIGHT avoids this by relying on local Hebbian updates $\Delta\mathbf{W}^{(l)} \propto (\mathbf{H}^{(l-1)})^\top\mathbf{R}^{(l)}$, which only requires forward states and local residuals. This significantly reduces both runtime constants and memory footprint.

SIGHT achieves comparable asymptotic complexity to conventional GNNs while avoiding the heavy memory burden of backpropagation. Event-driven sparsity further improves efficiency, enabling scalable, energy-aware training and inference for large graphs and safety-critical applications.

## C  THEORETICAL ANALYSIS

In this part, we provide detailed proofs for the theoretical claims in Section 3.6.

### C.1  CONVERGENCE OF PREDICTIVE CODING INFERENCE

Recall that the predictive coding (PC) energy function is defined as

$$\mathcal{F}(\mathbf{Z}) = \frac{1}{2} \sum_{l=1}^{L} \|\mathbf{Z}^{(l)} - f^{(l)}(\mathbf{H}^{(l-1)}; \mathbf{W}^{(l)})\|_2^2, \tag{16}$$

where $\mathbf{Z}^{(l)}$ is the latent representation of layer $l$, $f^{(l)}$ denotes the GNN message-passing operator with weights $\mathbf{W}^{(l)}$.

The inference dynamics follow the update rule:

$$\mathbf{Z}_{k+1}^{(l)} = \mathbf{Z}_k^{(l)} - \eta\, \varepsilon_k^{(l)}, \quad \varepsilon_k^{(l)} = \mathbf{Z}_k^{(l)} - f^{(l)}(\mathbf{H}^{(l-1)}; \mathbf{W}^{(l)}). \tag{17}$$

**Theorem 3** (Convergence of PC Inference). *Suppose each $f^{(l)}$ is Lipschitz continuous with constant $L_f < 1$. Then for any initialization $\mathbf{Z}_0^{(l)}$, the iterative dynamics converge linearly to a unique fixed point*

$$\mathbf{Z}^{(l)*} = f^{(l)}(\boldsymbol{H}^{(l-1)}; \boldsymbol{W}^{(l)}), \tag{18}$$

*which minimizes the local energy $\mathcal{E}(Z)$.*

*Proof.* We note that

$$\nabla_{\mathbf{Z}^{(l)}} \mathcal{F}(\mathbf{Z}) = \mathbf{Z}^{(l)} - f^{(l)}(\mathbf{H}^{(l-1)}; \mathbf{W}^{(l)}) = \varepsilon^{(l)}.$$

Thus, the PC update is equivalent to performing gradient descent on $\mathcal{F}(\mathbf{Z})$:

$$\mathbf{Z}_{k+1}^{(l)} = \mathbf{Z}_k^{(l)} - \eta \nabla_{\mathbf{Z}^{(l)}} \mathcal{F}(\mathbf{Z}).$$

For Lipschitz-smooth functions with $L_f < 1$, standard results in convex optimization guarantee convergence to a stationary point, provided $\eta < \frac{2}{L_f}$. Since $\mathcal{F}$ is quadratic in $\mathbf{Z}$, this stationary point is also the global minimizer. $\qquad\square$

### C.2  PREDICTIVE CODING RESIDUALS AS UNCERTAINTY ESTIMATORS

We define the predictive coding residual at node $i$ and layer $l$ as

$$u_i^{(l)} = \|\varepsilon_i^{(l)}\|_2. \tag{19}$$

**Proposition C.1.** *If the predictive function $f^{(l)}$ is well-calibrated on the in-distribution (ID) data, then larger residuals $u_i^{(l)}$ correspond to higher epistemic uncertainty, particularly under distribution shift.*

*Sketch.* In the ID case, residuals $\varepsilon^{(l)}$ are approximately zero-mean with bounded variance. Under OOD shift, the mismatch between the learned mapping $f^{(l)}$ and the new distribution causes systematic deviations in $\mathbf{Z}^{(l)}$, resulting in larger $\|\varepsilon^{(l)}\|$. Therefore, residual magnitudes can be used as a proxy for uncertainty. $\qquad\square$

#### C.2.1  HEBBIAN UPDATES AND GLOBAL RISK MINIMISATION

SIGHT updates weights locally via

$$\Delta \mathbf{W}^{(l)} \propto [\varepsilon^{(l)}]^\top \mathbf{H}^{(l-1)}. \tag{20}$$

**Lemma 1.** *The above local Hebbian update is equivalent, in expectation, to a stochastic gradient step on the global predictive risk*

$$\mathcal{R}(\boldsymbol{W}) = \mathbb{E}_{(X,y)} \Big[ \ell(f(\boldsymbol{X}; \boldsymbol{W}), y) \Big], \tag{21}$$

*where $\ell$ is the cross-entropy loss.*

*Sketch.* By Taylor expanding the cross-entropy loss around the current prediction, the gradient $\nabla_{\mathbf{W}^{(l)}}\ell$ can be decomposed into a local error term times the input activation $\mathbf{H}^{(l-1)}$. This aligns with the Hebbian update rule, showing that PC updates approximate the gradient of $\mathcal{R}(\mathbf{W})$ without backpropagation. $\qquad\square$

Theoretical analysis of SIGHT establishes three key results: the predictive coding inference dynamics are guaranteed to converge to local energy minima, ensuring stability of the learning process; the residual signals generated during inference naturally serve as interpretable measures of epistemic uncertainty, providing insights into model confidence; and the local Hebbian update rules approximate gradient descent on the global predictive risk, thereby enabling efficient learning without reliance on backpropagation.

# D  EXPERIMENTS

## D.1  DATASETS

According to Section 4.1, our experiments are conducted on five node classification datasets, Cora, Citeseer, Pubmed, Twitch, and CBAS. The Cora, Citeseer, and Pubmed citation networks contain nodes representing documents, edges representing citation links, bag-of-words feature vectors, and class labels corresponding to research topics. Since these datasets lack explicit domain information, we synthetically introduce covariate shifts by generating spurious node features. Concretely, we retain the original node labels and construct 6 domains (id $i \in \{1, 2, 3, 4, 5, 6\}$) with shifted features. For each domain, a randomly initialized GCN takes the node label $y_v$ and domain id $i$ as input to produce spurious features $\tilde{\mathbf{x}}_v^{(i)}$. The final node representation is obtained by concatenating the original and spurious features:

$$\mathbf{x}_v^{(i)} = \left[\mathbf{x}_v \,\|\, \tilde{\mathbf{x}}_v^{(i)}\right], \quad \mathbf{X}^{(i)} = \left[\mathbf{x}_v^{(i)}\right]_{v \in \mathcal{V}}. \tag{22}$$

We treat $\mathbf{X}^{(1)}$, $\mathbf{X}^{(2)}$, and $\mathbf{X}^{(3)}$ as ID data, while $\mathbf{X}^{(4)}$, $\mathbf{X}^{(5)}$, and $\mathbf{X}^{(6)}$ serve as OOD domains. This construction enables systematic evaluation of model generalisation under feature distribution shifts. The reported number of nodes in Table 5 corresponds to a single synthetic domain.

The Twitch dataset is a social graph where nodes represent users, edges represent friendship links, and features describe user metadata. The binary classification task aims to predict whether a user streams mature content. Following Gui et al. (2022), the domains are defined by user language, which ensures that the prediction target is not confounded by the language used. Finally, CBAS is a synthetic dataset adapted from BA-Shapes (Ying et al., 2019). It is constructed by attaching 80 house-shaped motifs to a 300-node Barabási–Albert base graph. The learning objective is a four-class node classification task, where each node is categorized as the top, middle, or bottom node of a house motif, or as part of the base graph. In place of constant node features, colored attributes are introduced, forcing OOD algorithms to address the spurious correlations between colors and labels under concept splits.

We formalize concept shift by viewing a dataset as a mixture of $|C|$ latent concepts $C = \{c_1, \ldots, c_{|C|}\}$. Let $P_{y_j, d_i}(\mathbf{Y})$ denote the output distribution that assigns probability one to value $y_j$ under domain $d_i$, i.e., $P(\mathbf{Y} = y_j \mid \mathbf{X}_{\text{ind}} = \mathbf{x}_i) = 1$. For the classification setting where $Y$ is discrete, the conditional distribution of a concept $c_k$ can be represented as a combination of one-dimensional distributions:

$$P_{c_k}(\mathbf{Y} \mid \mathbf{X}_{\text{ind}} = \mathbf{x}_i) = \sum_{j=1}^{|\mathcal{Y}|} q_{i,j}^k \, P_{y_j, d_i}(\mathbf{Y}), \tag{23}$$

where $q_{i,j}^k$ indicates the strength of the spurious correlation between domain $d_i$ and label $y_j$ in concept $c_k$. The overall dataset distribution can then be described as a mixture across concepts:

$$P(\mathbf{Y}, \mathbf{X}) = \sum_{k=1}^{|C|} w_k \, P_{c_k}(\mathbf{Y}, \mathbf{X}) = \sum_{k=1}^{|C|} w_k \, P(\mathbf{X})_{c_k} \, P_{c_k}(\mathbf{Y} \mid \mathbf{X}), \tag{24}$$

where $w_k$ denotes the mixture coefficient associated with concept $c_k$.

Table 5: Statistics of the experimental datasets.

| Dataset | # Nodes | # Edges | # Classes | # Features | Shift Type | Train/Val/ID Test/OOD Test(%) |
|---------|---------|---------|-----------|------------|------------|-------------------------------|
| Cora | 2,708 | 5,429 | 7 | 1,433 | Covariate | 25/12.5/12.5/50 |
| Citeseer | 3,327 | 4,732 | 6 | 3,703 | Covariate | 25/12.5/12.5/50 |
| Pubmed | 19,717 | 44,338 | 3 | 500 | Covariate | 25/12.5/12.5/50 |
| Twitch | 34120 | 892346 | 2 | 128 | Concept | 39.87/28.36/8.54/23.23 |
| CBAS | 700 | 3962 | 4 | 4 | Concept | 20/40/20/20 |

For all datasets, we follow the dataset splits in (Wu et al., 2024) and (Gui et al., 2022) for training, validation, ID and OOD testing. More details of the dataset statistics are shown in Table 5.

## D.2 BASELINES

**Post-hoc Calibration Models.** As introduced in section 4.1, we propose a variety of post-hoc strategies to calibrate model predictions. Their key advantage lies in flexibility, as they operate directly on the model's outputs without requiring any modification to the underlying architecture or training process. In our experiments, we adopt the post-hoc calibration models provided by Trivedi et al. (2024). Here is an introduction of these post-hoc strategies used in our experiments:

- CaGCN: (Wang et al., 2021) leverages the graph structure and an auxiliary GCN to generate node-wise temperatures.

- Dirichlet calibration: (Kull et al., 2019) models calibrated outputs with a Dirichlet distribution, capturing inter-class dependency in probability adjustment.

- Ensemble temperature scaling (ETS): (Zhang et al., 2020) extends this idea by combining multiple temperature-scaled models for improved flexibility.

- GATS: (Hsu et al., 2022) further incorporates graph attention to capture the influence of neighboring nodes when learning these temperatures.

- Multi-class isotonic regression (IRM): (Zhang et al., 2020) applies non-parametric isotonic regression to better capture non-linear calibration mappings.

- Order-invariant calibration: (Rahimi et al., 2020) enforces invariance to label permutations, ensuring consistent probability estimates across classes.

- Spline: Gupta et al. (2021) fits smooth spline functions to adjust predicted probabilities.

- Vector scaling (VS): (Guo et al., 2017) learns class-specific scaling parameters, allowing heterogeneous calibration across classes.

All code used in this work complies with the respective providers' licenses and does not include any personally identifiable information or offensive content. The repositories for the baseline implementations are listed below:

GCN (MIT license): https://github.com/tkipf/pygcn

GAT (MIT license): https://github.com/Diego999/pyGAT

G-$\Delta$UQ: https://github.com/pujacomputes/gduq/tree/main

## D.3 EVALUATION METRICS

According to section 4.1, we report five widely used metrics: Accuracy, Expected Calibration Error (ECE), Negative Log-Likelihood (NLL), Brier Score (BS), and the Area Under the Receiver Operating Characteristic Curve (AUROC). These metrics capture complementary aspects of accuracy and calibration.

**Accuracy.** Accuracy is the most widely used metric for evaluating node classification performance. It measures the proportion of correctly predicted labels over the total number of test nodes

and directly reflects the discriminative power of the model. Formally, given test set $\mathcal{D}_{\text{test}}$ with ground-truth labels $\{y_i\}$ and predicted labels $\{\hat{y}_i\}$, accuracy is defined as

$$\text{Accuracy} = \frac{1}{|\mathcal{D}_{\text{test}}|} \sum_{i \in \mathcal{D}_{\text{test}}} \mathbb{I}(\hat{y}_i = y_i), \tag{25}$$

where $\mathbb{I}(\cdot)$ denotes the indicator function. Higher accuracy indicates that the model has stronger classification capability, though it does not capture the quality of predictive probabilities or calibration, which motivates the use of additional uncertainty-aware metrics.

**Expected Calibration Error (ECE).** Calibration refers to the alignment between predicted confidence and empirical accuracy. Calibrated models are expected to generate confidence scores that accurately reflect the true likelihood of the predicted classes (Naeini et al., 2015; Guo et al., 2017; Ovadia et al., 2019). ECE partitions predictions into $M$ confidence bins $\{B_m\}_{m=1}^{M}$. For each bin, we compute the average confidence $\text{conf}(B_m)$ and the empirical accuracy $\text{acc}(B_m)$. ECE is defined as

$$\text{ECE} = \sum_{m=1}^{M} \frac{|B_m|}{n} \big| \text{acc}(B_m) - \text{conf}(B_m) \big|, \tag{26}$$

where $n$ is the total number of test samples. A smaller ECE indicates better calibration.

**Brier Score (BS).** The Brier Score measures the mean squared difference between predicted probability vectors and one-hot ground-truth labels:

$$\text{Brier} = \frac{1}{n} \sum_{i=1}^{n} \sum_{c=1}^{C} \big( p_\theta(y = c | x_i) - \mathbf{1}[y_i = c] \big)^2, \tag{27}$$

where $C$ is the number of classes and $\mathbf{1}[\cdot]$ denotes the indicator function. A lower Brier Score indicates that predicted probabilities are closer to the true distribution.

**Negative Log-Likelihood (NLL).** NLL evaluates the quality of probabilistic predictions. Given predicted probability distributions $p_\theta(y_i | x_i)$ for test samples $(x_i, y_i)$, it is defined as

$$\text{NLL} = -\frac{1}{n} \sum_{i=1}^{n} \log p_\theta(y_i | x_i). \tag{28}$$

Lower values correspond to higher likelihood assigned to the true labels, indicating better uncertainty modelling.

**Area Under ROC Curve (AUROC).** Given a classifier that outputs class probabilities $\{p_{i,c}\}_{c=1}^{K}$ (or logits $\{z_{i,c}\}_{c=1}^{K}$) for each sample $i$, let the hard prediction be

$$\hat{y}_i = argmax_c \, p_{i,c}, \qquad y_i \in \{1, \ldots, K\}. \tag{29}$$

Define the binary label for *error detection* as

$$y_i^{\text{bin}} = \mathbf{1}[\hat{y}_i \neq y_i] \in \{0, 1\}, \tag{30}$$

where 1 denotes a misclassification (positive class) and 0 denotes a correct prediction (negative class).

We compute an *uncertainty score* $s_i$ from the model outputs using MSP and entropy:

$$\text{MSP:} \quad s_i = 1 - \max_c p_{i,c}, \tag{31}$$

$$\text{Entropy:} \quad s_i = -\sum_{c=1}^{K} p_{i,c} \log p_{i,c}. \tag{32}$$

For a threshold $\tau$, predict "error" if $s_i > \tau$. The ROC statistics are

$$\text{TPR}(\tau) = \frac{\sum_i \mathbf{1}[s_i > \tau] \, \mathbf{1}[y_i^{\text{bin}} = 1]}{\sum_i \mathbf{1}[y_i^{\text{bin}} = 1]}, \tag{33}$$

$$\text{FPR}(\tau) = \frac{\sum_i \mathbf{1}[s_i > \tau] \, \mathbf{1}[y_i^{\text{bin}} = 0]}{\sum_i \mathbf{1}[y_i^{\text{bin}} = 0]}. \tag{34}$$

Table 6: Node classification accuracy and uncertainty calibration on ID and OOD datasets with the GAT backbone. Best results are in **bold**, and SIGHT is marked with ▢. Results with the GCN backbone are in Table 1.

| Method | Accuracy ↑ | | ECE ↓ | | NLL ↓ | | BS ↓ | | AUROC ↑ | |
|---|---|---|---|---|---|---|---|---|---|---|
| | ID | OOD | ID | OOD | ID | OOD | ID | OOD | ID | OOD |
| **Cora** | | | | | | | | | | |
| GAT | 90.6±0.5 | 67.4±4.1 | **0.015±0.001** | 0.164±0.026 | 0.286±0.027 | 1.072±0.108 | 0.137±0.011 | 0.498±0.059 | **88.9±0.7** | 71.1±3.2 |
| +G-ΔUQ | 93.9±0.4 | 80.2±3.5 | 0.015±0.002 | 0.061±0.033 | **0.211±0.007** | 0.578±0.098 | **0.097±0.004** | 0.291±0.053 | 87.2±1.6 | 81.8±3.0 |
| +SIGHT | **93.9±0.9** | **96.3±0.7** | 0.017±0.002 | **0.008±0.002** | 0.227±0.036 | **0.138±0.028** | 0.097±0.016 | **0.058±0.012** | 87.1±3.0 | **89.6±2.1** |
| **Citeseer** | | | | | | | | | | |
| GAT | 82.1±0.4 | 46.6±1.9 | 0.026±0.003 | 0.334±0.027 | 0.503±0.012 | 1.838±0.051 | 0.252±0.006 | 0.793±0.026 | **84.1±0.8** | 74.1±1.6 |
| +G-ΔUQ | 81.6±0.4 | 71.0±4.2 | 0.021±0.008 | 0.066±0.027 | 0.531±0.037 | 0.889±0.145 | 0.263±0.010 | 0.416±0.056 | 82.2±1.6 | 76.5±1.3 |
| +SIGHT | **88.8±0.5** | **93.4±0.2** | **0.020±0.006** | **0.028±0.003** | **0.414±0.022** | **0.261±0.012** | **0.175±0.010** | **0.105±0.005** | 77.7±1.74 | **80.8±1.2** |
| **Pubmed** | | | | | | | | | | |
| GAT | 88.0±0.1 | 73.7±1.9 | 0.008±0.001 | 0.146±0.028 | 0.306±0.001 | 0.960±0.081 | 0.173±0.000 | 0.411±0.028 | 85.2±0.3 | 72.3±1.5 |
| +G-ΔUQ | **91.3±0.2** | 83.8±0.7 | **0.006±0.001** | 0.090±0.009 | **0.236±0.002** | 0.668±0.048 | **0.129±0.002** | 0.263±0.013 | **86.3±0.4** | 74.2±0.3 |
| +SIGHT | 86.0±0.3 | **85.7±0.3** | 0.012±0.002 | **0.011±0.009** | 0.376±0.009 | **0.380±0.009** | 0.208±0.005 | **0.210±0.005** | 81.6±0.7 | **81.7±0.8** |
| **Twitch** | | | | | | | | | | |
| GAT | 60.7±9.0 | 57.5±4.1 | 0.129±0.066 | 0.046±0.028 | **0.685±0.006** | 0.683±0.007 | **0.492±0.006** | 0.490±0.008 | **59.5±4.3** | 49.2±3.3 |
| +G-ΔUQ | **64.8±9.9** | 58.3±2.9 | 0.163±0.067 | 0.065±0.024 | 0.690±0.021 | 0.687±0.009 | 0.497±0.021 | 0.494±0.009 | 58.0±9.1 | 45.4±1.5 |
| +SIGHT | 51.2±2.9 | **60.1±3.3** | **0.103±0.033** | **0.031±0.027** | 0.719±0.021 | **0.670±0.013** | 0.524±0.019 | **0.477±0.012** | 49.8±2.4 | **51.6±4.7** |
| **CBAS** | | | | | | | | | | |
| GAT | 75.7±1.9 | 70.4±2.7 | 0.104±0.034 | **0.112±0.022** | 0.593±0.042 | 0.837±0.074 | 0.324±0.026 | 0.429±0.040 | 81.5±6.3 | 70.4±2.8 |
| +G-ΔUQ | 80.1±6.8 | **73.0±7.2** | **0.101±0.030** | 0.123±0.047 | 0.529±0.128 | **0.759±0.134** | 0.284±0.072 | **0.389±0.082** | 82.1±2.3 | 74.8±8.1 |
| +SIGHT | **82.6±0.6** | 60.0±3.0 | 0.126±0.027 | 0.114±0.039 | **0.515±0.034** | 0.978±0.054 | **0.261±0.013** | 0.504±0.029 | **84.6±0.0** | **81.0±5.3** |

The *AUROC* is the area under this ROC curve:

$$\text{AUROC} = \int_0^1 \text{TPR}\big(\text{FPR}^{-1}(u)\big)\, du. \tag{35}$$

Intuitively, it measures how well $s_i$ ranks misclassified samples above correctly classified ones. The introduction of how threshold is selected is detailed in Appendix D.4.

## D.4 IMPLEMENTATION DETAILS

According to section 4.1, all experiments are conducted on a cloud server equipped with a single NVIDIA vGPU (48 GB memory) and 20 vCPUs (Intel Xeon Platinum 8470), with 90 GB system memory. The software environment includes Ubuntu 20.04, Python 3.8, PyTorch 2.0, PyTorch Geometric, snnTorch, and CUDA 11.8. We train models in a full-batch setting with Adam optimizer. For fair comparison, our framework and all baselines employ temperature scaling as a post-processing step for calibration.

As for the ROC thresholds $\tau$, they are not manually chosen but derived from the empirical distribution of uncertainty scores. Formally, let

$$\mathcal{S} = \{s_1, s_2, \ldots, s_n\}, \tag{36}$$

then the threshold set is

$$\mathcal{T} = \{-\infty\} \cup \mathcal{S} \cup \{+\infty\}. \tag{37}$$

That is, each observed uncertainty score (together with boundary points) is treated as a potential threshold, so AUROC evaluates performance across all possible thresholds rather than relying on a fixed one.

## D.5 ADDITIONAL RESULTS

### D.5.1 COMPARISON OF PERFORMANCE AND CALIBRATION

Table 6 reports node classification accuracy and calibration metrics under both ID and OOD settings with the GAT backbone. Overall, SIGHT consistently delivers competitive or superior performance compared to GAT and G-ΔUQ. Apart from CBAS, SIGHT achieves strong OOD accuracy while simultaneously reducing calibration errors, as reflected in the lowest OOD ECE and Brier scores. These improvements highlight the ability of predictive coding residuals to enhance uncertainty awareness beyond conventional GAT models. On the smaller CBAS datasets, SIGHT remains robust, matching or surpassing baselines in ID accuracy and achieving higher AUROC. Importantly, AUROC values confirm that SIGHT provides more reliable uncertainty estimation under distribution

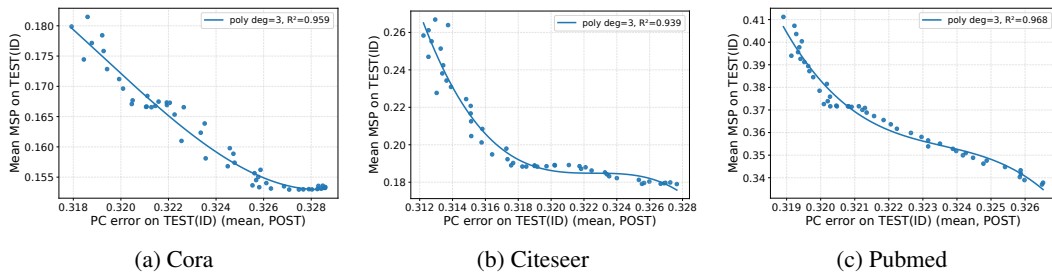

| (a) Cora | (b) Citeseer | (c) Pubmed |

Figure 3: Correlation between predictive coding (PC) error and confidence scores on the Cora, Citeseer, and Pubmed ID test sets. Higher PC errors consistently correspond to lower confidence, showing that PC residuals serve as an intrinsic measure of uncertainty.

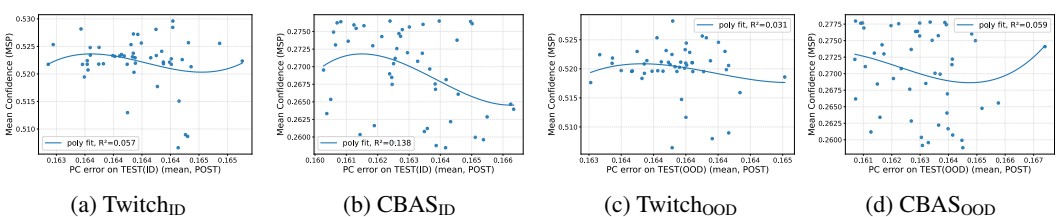

| (a) Twitch$_{ID}$ | (b) CBAS$_{ID}$ | (c) Twitch$_{OOD}$ | (d) CBAS$_{OOD}$ |

Figure 4: Correlation between predictive coding (PC) error and confidence scores on the Twitch and CBAS ID and OOD test sets. Higher PC errors correspond to lower confidence, indicating that PC residuals provide an intrinsic measure of uncertainty.

shifts, establishing a favorable balance between predictive accuracy and calibration quality across diverse domains.

### D.5.2 Explainability of Predictive Coding on Uncertainty Estimation

Other than OOD data shown in Figure 2, Figure 3 reports the correlation between PC error and confidence on the ID test sets of Cora, Citeseer, and Pubmed. Similar to the OOD case, we observe a strong negative correlation: higher PC errors correspond to lower confidence. This shows that predictive coding residuals not only capture local mismatches between predictions and inputs but also faithfully reflect model certainty on in-distribution samples. The high $R^2$ values of the polynomial fits (0.94–0.97) further confirm that PC errors provide reliable, interpretable indicators of confidence, supporting the claim that SIGHT yields uncertainty estimates without post-hoc calibration. In contrast, Figure 4 shows the correlation on Twitch and CBAS under both ID and OOD settings. Here, the correlations are weak (low $R^2$ values), suggesting that PC errors do not align well with confidence. This misalignment is particularly evident under concept shift, where disrupted label–feature correspondence causes predictive coding to propagate misleading error signals, undermining the reliability of uncertainty estimation. Overall, these results indicate that PC residuals are effective under covariate shift but limited under concept shift.

### D.5.3 Parameter Analysis

As introduced in 4.2, we further investigate the impact of different hyperparameters on SIGHT, including the number of predictive coding iterations $K$, the number of timesteps $T$, and the learning rate under both GCN and GAT backbones. Figure 5 presents the sensitivity analysis. Across datasets, performance remains stable across a wide range of $K$ and $T$, showing that SIGHT is robust to the choice of inference iterations and simulation length. In contrast, the learning rate has a more pronounced influence: overly large or small values lead to noticeable drops in AUROC, particularly on smaller datasets such as CBAS and Twitch. These findings demonstrate that SIGHT maintains strong and consistent performance without requiring fine-grained hyperparameter tuning, highlighting its practicality and robustness in real-world settings.

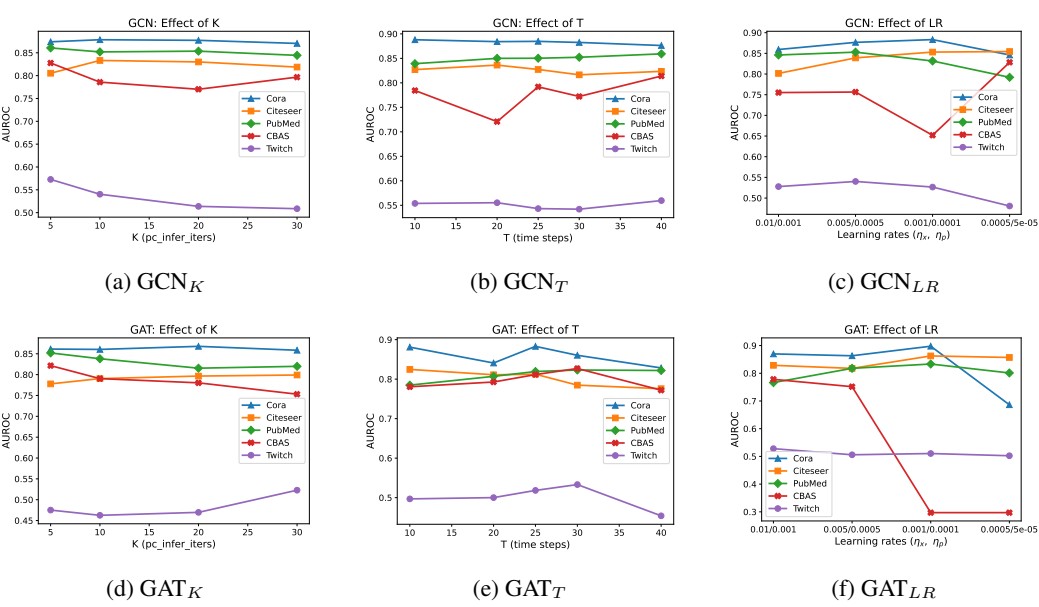

Figure 5: Sensitivity analysis of SGPC under different hyperparameters with GCN and GAT backbones. The effects of predictive coding iterations ($K$), timesteps ($T$), and learning rate ($LR$) on AUROC are evaluated across five datasets (Cora, Citeseer, Pubmed, CBAS, Twitch).

