# OpenReview forum: "Spiking Graph Predictive Coding"
_ICLR.cc/2026/Conference — ICLR 2026 Conference Withdrawn Submission_

### Official Review · Reviewer_EbQQ · 2025-10-17

**Soundness:** 2
**Presentation:** 3
**Contribution:** 2
**Rating:** 2
**Confidence:** 5

**Summary:**

In this paper, the authors focus on the calibration of GNNs under distribution shifts. They propose a framework named SIGHT that integrates SNNs with predictive coding to enhance OOD generalization and uncertainty calibration in GNNs. Specifically, authors replace traditional global backpropagation with a biologically inspired, local spike-driven error correction mechanism, allowing each layer to learn independently from its own predictive coding residuals. Extensive experiments are conducted on various graph learning benchmarks to verify the effectiveness of SIGHT.

**Strengths:**

Strengths:

- Clearly writing in the methodology part, so it is easy to follow the core of the proposed SIGHT.

- From the extensive experiments, the proposed SIGHT with GCN or GAT can consistently improve OOD generalization and uncertainty calibration across the selected graph learning benchmarks.

- The authors provide theoretical analyses in Appendix C to provide more intuition into the connection between predictive coding errors and OOD generalization bounds, as well as calibration error.

**Weaknesses:**

Weakness:

- In graph OOD settings, distribution shifts are typically induced by spurious or invariant features. However, this paper does not discuss these aspects nor provide experiments on widely used OOD benchmarks such as DrugOOD, IC50, or EC50 [1,2].
- In the abstract and introduction, the authors emphasize that SIGHT is **attractive for deployment on power-constrained hardware**. However, there are no experiments or evaluations conducted on any real or simulated low-power hardware platforms [3,4]. Moreover, since the core of this method lies in SNNs, whose essential motivation is event-driven, energy-efficient neural computation, experiments on neuromorphic or energy-sensitive datasets such as SEED and BCI are still missing [5,6].
- In experimental parts, no recent Post-hoc Calibration methods are deployed for fair comparisons.
- In Algorithm 1, the authors provide the training process of SIGHT. However, it consists of $L$ layers message passing and $T$ time steps spiking processing, resulting in $K \times T$ repeated computations. This substantially increases training time and computational cost, contradicting the claimed efficiency advantages of SNNs.

[1] Empowering Graph Invariance Learning with Deep Spurious Infomax.

[2] Disentangling invariant subgraph via variance contrastive estimation under distribution shifts.

[3] Spiking graph convolutional networks.

[4] Scaling up dynamic graph representation learning via spiking neural networks.

[5] Differential entropy feature for eeg-based emotion classification.

[6] Bci competition 2008–graz data set.

**Questions:**

Questions:

- The local Hebbian update rule ($\Delta \mathbf{W}^{(l)} \propto (\mathbf{H}^{(l-1)})^T \mathbf{R}^{(l)}$) in Appendix.C is only heuristically justified as "being equivalent to a stochastic gradient step". So, why can this **equivalent** be achieved?
- Predictive coding has been proposed previously. What are the key differences between [7] and the proposed SIGHT framework, apart from the application scenarios?

[7] Predictive coding with spiking neurons and feedforward gist signaling.

---

### Official Review · Reviewer_XV4X · 2025-10-31

**Soundness:** 2
**Presentation:** 2
**Contribution:** 2
**Rating:** 2
**Confidence:** 3

**Summary:**

This paper introduces SpIking GrapH predicTive coding (SIGHT), a biologically inspired framework that integrates spiking neural computation with predictive coding (PC) for graph representation learning. The model replaces global backpropagation with local Hebbian-style updates and iterative PC inference, aiming to improve uncertainty calibration and out-of-distribution (OOD) generalization in graph neural networks. The study provides theoretical analyses linking predictive coding error to OOD generalization and calibration error bounds. In experiments, SIGHT is performed on five benchmark datasets under both covariate and concept shift settings. Empirical results show that SIGHT consistently improves calibration metrics (e.g., ECE, NLL and BS) and OOD detection performance compared to strong baselines such as G-$\Delta$UQ and post-hoc calibration models.

**Strengths:**

1. The combination of predictive coding with spiking dynamics is innovative. It offers a novel alternative to backpropagation-based optimization for graph learning.

2. The study presents a thorough experimental evaluation under both covariate and concept shifts. It provides diverse evaluation metrics across multiple datasets to demonstrate the effectiveness of the proposed method.

**Weaknesses:**

1. Most of the theoretical results are adapted from prior works on domain adaptation and predictive coding, but relevant references are missing in the theoretical analysis section. Moreover, the connection between the proposed theorems and the spiking dynamics is weak. None of the theoretical analyses explicitly depends on spiking mechanisms. The claimed theoretical contributions seem only loosely related to the proposed model.

2. As the core innovation, the spiking-based predictive-coding loop in Equation 3 lacks the theoretical support or explanations. Appendix B.2 only describes standard LIF neuron dynamics. A deeper discussion or comparison to existing graph predictive-coding methods would strengthen the contribution.

3. In Table 1, GCN+SIGHT significantly outperforms the full-precision GCN on most ID datasets. Such significant performance gains are atypical for spiking GNNs that usually trade a small amount of accuracy for efficiency. The paper should provide more explanations for why the low-precision, event-driven model surpasses full-precision baselines to this extent. It raises my concern about evaluation consistency or fairness.

4. Although the paper claims that the experimental setting stems from a previous benchmark [1], the description in Appendix D.1 seems to be inconsistent with the artificial distribution shift reported in the benchmark. To substantiate that the reported accuracy improvements stem from the proposed method rather than the non-standard experimental setting, it is imperative to provide more explanations or necessary preliminaries.

[1] Handling distribution shifts on graphs: An invariance perspective, 2022.

**Questions:**

Please see the weaknesses.

---

### Official Review · Reviewer_uLVe · 2025-10-31

**Soundness:** 3
**Presentation:** 3
**Contribution:** 2
**Rating:** 4
**Confidence:** 3

**Summary:**

This paper proposes SIGHT (Spiking Graph Predictive Coding), a framework that integrates predictive coding dynamics with spiking neural computation for robust graph learning. By combining local Hebbian-style weight updates and event-driven spiking computation, SIGHT replaces global backpropagation with local, biologically plausible error correction and uncertainty quantification mechanisms. The paper provides theoretical analysis on convergence and out-of-distribution (OOD) generalization capability, and evaluates the method on five graph datasets involving covariate and concept shift scenarios. Experiments demonstrate that SIGHT outperforms standard GNNs and uncertainty quantification baselines in terms of prediction accuracy, model calibration, and OOD detection.

**Strengths:**

1. The theoretical section provides analytical results, with Theorem 1 (OOD generalization bound) and Theorem 2 (convergence/ECE bound), establishing connections between predictive coding residuals and generalization performance as well as calibration guarantees, thereby strengthening the method's formal foundation.

2. Comprehensive experiments on five datasets cover both covariate and concept shift scenarios, supporting its superiority over GCN/GAT baselines in terms of OOD robustness and uncertainty calibration; ablation studies are thorough.

3. Well-written and well-organized, with a clear paper structure and a well-designed model illustration.

**Weaknesses:**

1. The paper primarily combines existing elements (LIF spiking neurons, Hebbian-style learning and local error feedback) and applies them to graph datasets, thus lacking sufficient novelty.

2. The "Hebbian update" is only briefly mentioned as the "outer product of presynaptic firing rate and postsynaptic residual." A more detailed explanation is missing, including how the cross-entropy gradient is approximated.

3. The comparison with baseline models is insufficient. The paper only compares against a single baseline method, which is inadequate to fully validate the performance advantages of the proposed approach.

4. In Section 4.1, "recognizee" should be "recognized".

**Questions:**

1. Could the authors further clarify the complete dynamic process from continuous predictive coding errors to discrete spike generation and synaptic updates? The mechanism describing how prediction errors drive neuronal spiking and subsequently trigger local synaptic updates is not sufficiently clear.

2. Why can spiking activity be naturally integrated into the predictive coding framework? What advantages does this integration offer—compared to using spiking networks or predictive coding alone—in terms of information representation or robustness?

3. What are the actual training and inference times and memory consumption of SIGHT compared to standard GNNs? Especially as dataset size increases, could the authors provide practical runtime comparisons with standard GNNs or methods such as G-ΔUQ?

---

### Note · Authors · 2025-11-12

I have read and agree with the venue's withdrawal policy on behalf of myself and my co-authors.